# Impaired Function of PLEKHG2, a Rho-Guanine Nucleotide-Exchange Factor, Disrupts Corticogenesis in Neurodevelopmental Phenotypes

**DOI:** 10.3390/cells11040696

**Published:** 2022-02-16

**Authors:** Masashi Nishikawa, Hidenori Ito, Hidenori Tabata, Hiroshi Ueda, Koh-ichi Nagata

**Affiliations:** 1Department of Molecular Neurobiology, Institute for Developmental Research, Aichi Developmental Disability Center, 713-8 Kamiya, Kasugai 480-0392, Japan; mnishikawa@inst-hsc.jp (M.N.); itohide@inst-hsc.jp (H.I.); tabata@inst-hsc.jp (H.T.); 2United Graduate School of Drug Discovery and Medical Information Sciences, Gifu University, Yanagido 1-1, Gifu 501-1193, Japan; hueda@gifu-u.ac.jp; 3Department of Neurochemistry, Nagoya University Graduate School of Medicine, 65 Tsurumai-cho, Nagoya 466-8550, Japan

**Keywords:** PLEKHG2, Rho family small GTPases, RhoGEF, brain development, axon guidance, dendrite formation, dendritic spine development

## Abstract

Homozygosity of the p.Arg204Trp variation in the Pleckstrin homology and RhoGEF domain containing G2 (*PLEKHG2*) gene, which encodes a Rho family-specific guanine nucleotide-exchange factor, is responsible for microcephaly with intellectual disability. However, the role of PLEKHG2 during neurodevelopment remains unknown. In this study, we analyzed mouse Plekhg2 function during cortical development, both in vitro and in vivo. The p.Arg200Trp variant in mouse (Plekhg2-RW), which corresponds to the p.Arg204Trp variant in humans, showed decreased guanine nucleotide-exchange activity for Rac1, Rac3, and Cdc42. Acute knockdown of Plekhg2 using in utero electroporation-mediated gene transfer did not affect the migration of excitatory neurons during corticogenesis. On the other hand, silencing Plekhg2 expression delayed dendritic arbor formation at postnatal day 7 (P7), perhaps because of impaired Rac/Cdc42 and p21-activated kinase 1 signaling pathways. This phenotype was rescued by expressing an RNAi-resistant version of wildtype Plekhg2, but not of Plekhg2-RW. Axon pathfinding was also impaired in vitro and in vivo in Plekhg2-deficient cortical neurons. At P14, knockdown of Plekhg2 was observed to cause defects in dendritic spine morphology formation. Collectively, these results strongly suggest that PLEKHG2 has essential roles in the maturation of axon, dendrites, and spines. Moreover, impairment of PLEKHG2 function is most likely to cause defects in neuronal functions that lead to neurodevelopmental disorders.

## 1. Introduction

Rho family guanosine triphosphatases (GTPases) are known to regulate various cellular processes, such as morphology, gene transcription, proliferation, and migration through actin cytoskeletal rearrangement, and they play crucial roles in various cell types [1,2,3,4]. Similarly to other GTPases, Rho family proteins are binary switches that cycle between an inactive (GDP-bound) and an active (GTP-bound) conformational state. Interconversion of the two forms is controlled in a variety of ways. In general, guanine nucleotide exchange factors (GEFs) promote GDP/GTP exchange to activate Rho GTPases, which then initiate recognizing downstream effectors [5,6,7]. PLEKHG2 (Pleckstrin homology and RhoGEF domain containing G2), which is a GEF for Rac and Cdc42, interacts with Gβγ subunits of heterotrimeric G proteins and is activated upon co-expression of Gβγ in various cell types; whether activation is mediated directly by Gβγ remains to be clarified [8,9,10,11,12]. Rac and Cdc42 are crucial for the spatiotemporal activation of various effectors to control cell morphology through cytoskeletal organization, and they have been reported to be important for the cortical neuron development [13,14,15,16,17,18].

Recently, whole-exome sequence analysis has identified a homozygous missense variation resulting in, c.610C > T/p.Arg204Trp, in the PLEKHG2 protein of five patients with profound intellectual disability (ID), dystonia, postnatal microcephaly, and a distinctly abnormal neuroimaging pattern [19]. Notably, the missense variation is located in the Dbl homology (DH) domain that is responsible for activating the GTPases by catalyzing the exchange of GDP for GTP. Thus, PLEKHG2-Rac/Cdc42 signaling is likely to play an essential role in a variety of neuronal functions and, when defective, is involved in neurodevelopmental and psychiatric disorders. However, the precise role of PLEKHG2 in the neurodevelopmental process is unknown.

In this study, we performed both in vitro and in vivo analyses to elucidate the role of Plekhg2 protein during mouse corticogenesis, including excitatory neuron migration, dendritic arbor formation, axon elongation, and spine formation. Our results indicate that PLEKHG2 has essential roles in the maturation of axons, dendrites, and spines. Moreover, impairment of its function should cause defects that lead to neurodevelopmental disorders.

## 2. Materials and Methods

### 2.1. Ethics Statement

We followed the fundamental guidelines for proper conduct of animal experiments and related activities at academic research institutions under the jurisdiction of the Ministry of Education, Culture, Sports, Science, and Technology (Tokyo, Japan). All protocols for animal handling and treatment were reviewed and approved by the Animal Care and Use Committee of Institute for Developmental Research, Aichi Developmental Disability Center (approval number: 2019-013).

### 2.2. Plasmids

Mouse Plekhg2 cDNA was purchased from Kazusa DNA Research Institute and constructed into pCAG-Myc vector. Using wild-type *Plekhg2* (Plek2-WT) as a template, site-directed mutagenesis was performed using the KOD-Plus Mutagenesis kit (Toyobo Inc., Tokyo, Japan) to generate the *Plekhg2* variant p.Arg200Trp (Plek2-RW) corresponding to p.Arg204Trp in *PLEKHG2*. The following target sequence in *Plekhg2* was inserted into pSuper-puro RNAi vector (Cat# VEC-PBS-0008, OligoEngine, Seattle, WA, USA): Plek2#1, GGAAAACAGCCTACATTGT (1269–1287) and Plek2#2, GCAACTTCCAGAACATACA (2795–2813). Numbers indicate the positions from translational start sites. These RNAi vectors were termed sh-Plek2#1 and sh-Plek2#2. For RNAi-resistant versions of Plek2-WT and Plek2-RW, 4 silent variations were introduced in the target sequence as underlined (GCAATTTTCAAAATATACA in Plek2#2) and termed as Plek2-WT-R and Plek2-RW-R. As a control RNAi vector, we used sh-Luc designed against luciferase, CGTACGCGGAATACTTCGA (155–173) [20]. Mouse *Rac1*, *Rac3*, and *Cdc42* were amplified by PCR from a cDNA pool of mouse brain and cloned into pCAG-Myc vector. pF5A-CMV-neo-Gβ_1_ and -Gγ_2_ were produced as described previously [8]. pRK5-Myc-PAK1 (p21-activated kinase 1) and -PAK1LF, a kinase-active variant with a single amino acid substitution p.L107F, were kindly provided by the late Prof. Alan Hall (Univ. College London, UK), and subcloned into pCAG-Myc vector. p21-binding domain (PBD) in human PAK1 (aa 67–150) was amplified by PCR from a cDNA pool of U251MG cells and constructed into pGS21a vector (GenScript, Piscataway, NJ, USA). All constructs were verified by DNA sequencing.

### 2.3. Antibodies and Histochemical Reagents

The following antibodies were used: anti-GFP (Medical & Biological Laboratories, Nagoya, Japan, Cat# 598, RRID: AB_591819 or Nacalai Tesque, Kyoto, Japan, Cat# 04404-84, RRID: AB_10013361), anti-Myc (Medical & Biological Laboratories, Nagoya, Japan, Cat# M047-3, RRID: AB_591112), and anti-phospho-PAK1 (Thr423) (Cell Signaling Technology Cat#2601, RRID: AB_330220). Alexa Fluor 488 (Invitrogen, Carlsbad, CA, USA) was used as a secondary antibody. 4′, 6-diamidino-2-phenylindole (DAPI; Nichirei Bioscience, Tokyo, Japan) was used for staining DNA.

### 2.4. Cell Culture and Transfection

COS7 cells, i.e., monkey kidney fibroblast-like cells, and primary cortical neurons derived from embryonic day (E) 16.5 mice were cultured as described [20]. Transient transfections were carried out using polyethyleneimine “MAX” reagent (for COS7 cells) (Polysciences Inc., Warrington, PA, USA) or Neon transfection system (for primary neurons) (Invitrogen, Carlsbad, CA, USA).

### 2.5. Pull-Down Assay

Glutathione S-transferase (GST)-fused PBD of PAK1 was expressed in Escherichia coli BL21 (DE3) strain. Recombinant GST-protein was purified according to the manufacturer’s instructions. COS7 cells were transfected with pCAG-Myc-Plek2-WT, -Plek2-RW, -Rac1, -Rac3, -Cdc42, pF5A-CMV-neo-Gβ_1_, and -Gγ_2_ (0.2 μg/30 mm-dish) in various combinations. After 24 h of transfection, cells were serum-starved for 24 h. Transfected cells were then washed once with ice-cold PBS, and lysed with the pull-down buffer (50 mM Tris-HCl, pH7.5, 150 mM NaCl, 5 mM MgCl_2_, 0.1% SDS, 1% Nonidet P-40, and 0.5% deoxycholate), and insoluble materials were removed by centrifugation. The supernatant was then incubated for 30 min at 4℃ with Glutathione Sepharose 4B beads (GE Healthcare Life Sciences, Buckinghamshire, England) to which GST-fused PBD was bound. The beads were washed twice with the pull-down buffer. Bound proteins were solubilized in SDS-sample loading buffer and analyzed by Western blotting. Images were captured with LAS-4000 luminescent image analyzer (GE Healthcare Life Sciences).

### 2.6. In Utero Electroporation

Three-month-old specific-pathogen-free ICR pregnant mice (Japan SLC, Shizuoka, Japan) were housed singly in cage (197 mm × 340 mm × 165 mm) under a 12 h light-dark cycle with free access to autoclaved food (CE-2, CLEA Japan, Inc., Tokyo, Japan). Surgery on pregnant mice and embryo manipulations in the uterus were performed as previously described [21]. At E14, pregnant mice (Japan SLC, Shizuoka, Japan) were deeply anesthetized with a mixture of medetomidine (0.75 mg/kg), midazolam (4 mg/kg), and butorphanol (5 mg/kg). In utero electroporation was then performed in specific-pathogen-free animal facilities as described [22,23]. Briefly, the indicated amounts of pCAG expression and RNAi vectors were injected in various combinations with pCAG-EGFP (0.5 μg each) into the lateral ventricle of embryos using a glass micropipette made from a microcapillary tube (GD-1; Narishige, Tokyo, Japan). Then, each embryo was placed between a tweezer-type disc electrode (5-mm in diameter) (CUY650-5; NEPA Gene, Chiba, Japan) and subjected to five electronic pulses (50 ms of 35 V) at 450 ms intervals using an electroporator (NEPA21; NEPA Gene). In this way, plasmids were introduced into the somatosensory area, which is part of the parietal lobe. Brains were fixed at indicated postnatal day, sectioned, and then analyzed. All experimental procedures were carried out during day time. No animals were excluded, and none died during experimentation.

### 2.7. Time-Lapse Imaging

Transfected neurons were cultured in an incubator chamber (5% CO_2_ and 40% O_2_, at 37 °C) fitted onto an A1R HD25 confocal laser microscope (Nikon, Tokyo, Japan). Approximately 1–15 optical Z sections were acquired automatically every 1 min for 1 h, and about 10 focal planes (~30 µm-thickness) were merged to visualize the entire shape of the cells.

### 2.8. Immunofluorescence

Immunofluorescence analysis was conducted essentially as described [21]. As for cortical slice staining, brains were embedded in 3% agarose, cut into sections (100 μm-thickness) using a vibratome, and photographed with an LSM-880 confocal laser microscope (Carl Zeiss, Oberkochen, Germany). Acquired images were analyzed using ImageJ to determine cell morphology and fluorescence intensity or using NeuronStudio to quantify dendritic spine density [24].

### 2.9. Statistical Analysis

For all cell imaging experiments, counting and traces were assessed in a blinded manner. Statistical significance for multiple comparisons was determined by Dunnett’s or Tukey’s test (https://cran.r-project.org/web/packages/multcomp/multcomp.pdf (accessed on 8 January 2022). Comparisons between two groups were performed using Welch’s *t*-test [25]. *p* < 0.05 was considered statistically significant. Statistical analyses were performed using R (https://intro2r.com/citing-r.html; https://cran.r-project.org/doc/FAQ/R-FAQ.html#Citing-R (accessed on 8 January 2022).

## 3. Results

### 3.1. Characterization of Plekhg2 Variation

Since the p.Arg204Trp missense variation in human PLEKHG2 resides within the DH domain, which is responsible for activating Rac/Cdc42 by catalyzing the exchange of GDP for GTP, PLEKHG2 harboring this variation is likely to manifest abnormal GEF activity. We, therefore, examined whether this variation affects the GEF activity for Rac1, Rac3, and Cdc42, which have been reported to be important for the cortical neuron development. To this end, we utilized a COS7-cell transient expression system, in which Myc-tagged small GTPases and Gβγ were expressed in various combinations. When compared with wild-type mouse Plekhg2, i.e., Plek2-WT, the basal GEF-activity of Plekhg2 p.Arg200Trp, i.e., Plek2-RW, which is the mouse equivalent to human PLEKHG2 p.Arg204Trp, was only weakly affected (Figure 1A–C, lane 1 and 3). Although co-expression with Gβγ enhanced GEF-activity of Plek2-WT as previously reported [8] (Figure 1A–C, lane 1 and 2), Gβγ-mediated augmentation of Plek2-RW activity was decreased for the GTPases tested (Figure 1A–C, lane 2 and 4). Similarly, when PAK1 activity was assessed by T423-phosphorylation of Myc-PAK1 expressed in COS7 cells, Gβγ-stimulated Plek2-RW significantly decreased the phosphorylation when compared to Plek2-WT [26] (Figure 1D). Taken together, we assume that the disease-causative p.Arg204Trp variation impairs Gβγ-stimulated catalytic activity of PLEKHG2 on RAC and CDC42. However, further intensive in vivo examination is essential to determine whether this signaling defects indeed reflect the physiologically relevant substrate specificity of PLEKHG2.

### 3.2. Role of Plekhg2 in the Dendritic Arbor Development In Vivo

Rac/Cdc42-PAK1 signaling is most likely to be impaired in the cells expressing Plek2-RW from the results in Figure 1. Since the Rac/Cdc42-PAK1 signaling has been reported to be crucial for brain development [27,28,29,30,31], we analyzed the role of PLEKHG2 in corticogenesis by acute knockdown with an in utero electroporation method. To this end, we designed 2 RNAi vectors, sh-Plek2#1 and #2, against distinct regions in the mouse *Plekhg2* coding sequence. First, we examined the role of Plekhg2 in the migration of newly generated cortical neurons, because organized neuronal migration is essential for brain development. pCAG-GFP was co-electroporated with 1.0 μg of sh-Luc (control), sh-Plek2#1, or sh-Plek2#2 into the ventricular zone (VZ) progenitor cells of E14.5 brains. When localization of transfected cells and their progeny was analyzed at P0, neurons were found to migrate normally to layers II–III of the cortical plate in the control and knockdown experiments (Figure 2A), indicating that Plekhg2-knockdown has little effects on cortical neuron migration.

Given that aberrant synaptic network formation is linked to defective brain development and function, Plekhg2 may be involved in dendrite and/or axon growth during corticogenesis. When we examined the role of Plekhg2 in dendritic arbor formation, introduction of sh-Plek2#1 and #2 (1.0 μg) into the VZ cells at E14.5 resulted in highly suppressed dendritic arborization at P7 (Figure 2B). Branch point number was significantly reduced in the deficient neurons compared to that in matching control cells, indicating that Plekhg2 participates in dendritic arbor formation (Figure 2C). Rescue experiments were then conducted to rule out off-target effects. We prepared RNAi-resistant versions of wild type Plekhg2 and the variant, Plek2-WT-R and Plek2-RW-R, respectively, and confirmed that they were resistant to sh-Plek2#2 in COS7 cells (Figure 2D). Although the suppressed dendritic arborization by silencing Plekhg2 was rescued by Plek2-WT-R, Plek2-RW-R could not rescue this abnormal phenotype (Figure 2E,F). These results demonstrate the pathophysiological significance of the variation.

Based on the obtained results in Figure 1, it is possible that the dendritic arbor development is regulated by Plekhg2-Rac/Cdc42-PAK1 signaling. We here focused on the roles of Rac3 and Cdc42, which were previously reported to be important for the development of cortical neurons [14]. Co-expression of wild type Rac3 and Cdc42, perhaps due to the increased basal activity in neurons, rescued the phenotype of Plekhg2-knockdown in terms of the branch number of dendrites (Figure 3A,B). It should be noted that neuronal migration was not affected under the experimental conditions where 0.1 μg of wild type Rac3 or Cdc42 was electroporated. On the other hand, rescue effects by Rac1 were not characterized since simple overexpression of wildtype Rac1 impaired neuronal migration on its own (data not shown). Likewise, PAK1LF, an activated variant of PAK1, also rescued the phenotype (Figure 3C,D). Given the crucial role of Rac/Cdc42-PAK1 signaling in actin cytoskeletal reorganization, impaired regulation of Plekhg2 should dysregulate actin dynamics and disrupt the function of the actin cytoskeleton, leading to defects in dendritic development.

### 3.3. Role of Plekhg2 in the Dendritic Spine Formation

Since Plekhg2 was shown to be involved in dendrite network formation, we further explored whether Plekhg2 participates in the synapse formation and/or maintenance. It is also notable that ID, the main clinical features of the individuals with p.Arg204Trp in *PLEKHG2*, is associated with synaptic dysfunction. To this end, we assessed dendritic spine formation in cortical neurons following Plekhg2-knockdown in vivo. Notably, although Plekhg2-deficient neurons were differentiated with numerous dendrites at P14, the number appeared to be less than the control neurons (Figure 4A). Under the conditions, spine density was significantly decreased in neurons transfected with sh-Plek2#2, and the phenotype was rescued by Plek2-WT-R, but not Plek2-RW-R (Figure 4B,C). On the other hand, knockdown of Plekhg2 little affected the spine length and head diameter (Appendix A). Further analyses are required to determine if the observed impaired spine formation is a primary phenotype or secondary to dendrite growth defects at this timing. In any case, the functional disruption in Plekhg2 is most likely to impair synaptic connectivity through the defects in dendritic spine morphology. The pathophysiology of ID with *PLEKHG2* gene abnormalities may reflect these observed cellular phenotypes.

### 3.4. Role of Plekhg2 in the Axon Growth In Vitro and In Vivo

As a next set of experiments, we examined the role of Plekhg2 in the axon growth. When control sh-Luc was introduced into primary cultured cortical neurons at div0 and analyzed at div2, cells were normally differentiated in a time-dependent manner (Figure 5A,B, and Appendix A). In the meantime, Plekhg2-deficient neurons also appeared to be similarly differentiated in terms of axon elongation at this timing, perhaps due to the time-lag of the knockdown effects. Drastic phenotype was, however, observed when looking closer; the axonal growth cone of the deficient neurons was disrupted (Figure 5C,D, and Appendix A). The axon growth velocity of Plekhg2-silenced neurons was severely reduced when compared to control cells at div2 (Figure 5E), suggesting that Plekhg2 is important for the growth cone-mediated axon guidance. We further analyzed the physiological significance of Plekhg2 in interhemispheric axon projections of cortical neurons in vivo. When Plekhg2 was silenced in the VZ cells at E14.5, axon bundles extended normally into the contralateral hemisphere at P7 (Appendix A). However, such axons did not extend properly into the cortical layers on the contralateral side after reaching the contralateral white matter at P7 (Figure 5F). These results suggest that Plekhg2 is involved in the axon pathfinding of cortical neurons.

## 4. Discussion

De novo *RAC1* missense variants have been identified in neurodevelopmental disorders with global developmental delay/ID and brain size abnormalities as core phenotypes (Mental Retardation autosomal dominant 48, MRD48, OMIM #617751) [13]. The striking correlation between RAC1 activation levels and the head size of the affected individuals has been shown; the active and inactive variants tend to be associated with microcephaly and macrocephaly, respectively [13]. Interestingly, microcephaly and macrocephaly also have been described in individuals harboring distinct variants in TRIO, a RAC1-GEF [32,33,34]. Considering the relationship of microcephaly with the PLEKHG2 variation, impaired RAC1-signaling might be partially related to the clinical features of the patients. The present study showed that Plekhg2 plays essential roles in dendritic arbor development, axon pathfinding, and synapse formation/maintenance of cortical neurons during cortical development. It should be here noted that, unlike TRIO, Plekhg2 is not involved in neuronal migration (Figure 2A), suggesting that Plekhg2 appears to spatiotemporally activate Rac1 which functions in the “cell morphology machinery” rather than the “migration machinery”. Given many RhoGEFs and RhoGTPases expressed in neurons [35,36], a variety of RhoGEFs including PLEKHG2 should coordinatedly modulate various Rho-mediated cellular events in spatiotemporally regulated manners during brain development.

The p.Arg204Trp variation in PLEKHG2 is reported to cause “postnatal” microcephaly; head circumference of the patients were normal at birth and fell in percentiles to approximately −3 S.D. at 4–6 months of age [19]. In addition, MRI analyses of the patients at 8 months to 5 years of age disclosed abnormal white matter with a diffuse high signal on T2 and atrophy apparently due to paucity of white matter [19]. Considering that Plekhg2-knockdown had little effect on neuronal precursor proliferation (Appendix A), the postnatal microcephaly associated with the *PLEKHG2* variation is supposed to be caused by defects in neurite growth after birth. On the other hand, given that ID is referred to as a “synaptic” disorder, impaired synaptic function due to defects in dendritic as well as spine development may underlie the pathogenesis and pathophysiology of ID of the patients.

In utero electroporation-based analyses confirmed that Plekhg2-knockdown had little effects on axon extension of cortical neurons, but suppressed axon pathfinding after reaching contralateral cortex (Figure 5F). Additionally, Plekhg2-knockdown in dissociated cortical neurons induced axonal growth cone collapse (Figure 5A–E). Although the apparently normal phenotype of the axon extension observed in vitro may be due to the time-lag of the short hairpin RNA expression, Plekhg2-mediated spatiotemporal activation of Rac/Cdc42 is possible to regulate axon pathfinding through the control of the growth cone rather than axon elongation.

It is notable that the International Mouse Phenotyping Consortium has reported that the *Plekhg2*-knockout mice displayed only decreased body length out of 75 parameters assessed (https://www.mousephenotype.org/data/genes/MGI:2141874 (accessed on 12 January 2022). On the other hand, we here observed that acute knockdown of Plekhg2 is involved in cortical architecture. We assume that conditional and acute knockdown by the combination of in utero electroporation and RNAi circumvents the compensatory effects of general gene-knockout approaches. Given a variety of GEFs for Rac and Cdc42 expressed in developing central nervous system, it is possible that yet unidentified GEF(s) may compensate for the function of Plekhg2 in the *Plekhg2*-knockout mice.

As for the molecular mechanism underlying the PLEKHG2-deficient phenotypes, it remains to be clarified whether coordinated interaction of PLEKHG2 with Gβγ is pivotal for brain development. In this context, it is notable that Arg204 of PLEKHG2 is positioned in DH domain which is responsible for interacting with both Rac/Cdc42 and Gβγ [8], indicating possible influence by the p.Arg204Trp variation on the interaction of PLEKHG2 with Gβγ. Indeed, p.Arg200Trp variation was shown to abrogate the ability of Gβγ to activate Plekhg2 (Figure 1A–C) and PAK1 (Figure 1D) in COS7 cells. Considering that Gβγ is released from the heterotrimeric G proteins in responses to cell surface receptor stimulation and is associated with neurodevelopmental disorders [37,38,39], receptor-mediated signaling might play a role in neuronal morphology and synaptic plasticity. Further investigation is required to clarify the physiological significance of Gβγ-dependent Plekhg2 GEF activity for Rac/Cdc42 and subsequent PAK1 activation.

Although the spine structure is regulated by various molecular mechanisms, actin filaments provide the main foundation for spine shape, motility, and stability. We have previously reported that Plekhg2 localizes at both excitatory and inhibitory synapses [40], suggesting an essential role of Plekhg2 in synapse formation/maintenance. We assume that Plekhg2-Rac/Cdc42-PAK1 signaling also takes part in spine formation in vivo and reduced spine density observed in Figure 4 may be a primary phenotype. Indeed, the p.Arg200Trp variation in *Plekhg2* was found to be essential for the impaired spine formation since reduced spine density of cortical neurons by Plekhg2-knockdown was rescued by wild type Plekhg2 but not by the variant (Figure 4). To elucidate the physiological and pathophysiological significance of Plekhg2 in spine formation, identification of the spatiotemporal signaling network involving Plekhg2 will be essential.

In the present study, we clarified that PLEKHG2 is crucial for brain development based on the results that Plekhg2-deficiency caused morphological defects in cortical neurons during brain development. Additionally, the p.Arg204Trp variation in *PLEKHG2* is known to cause neurodevelopmental disorders with microcephaly and ID. As for an underlying mechanism in the clinical phenotypes, disrupted actin regulation caused by aberrant Rac/Cdc42-PAK1 signaling was supposed to be a core event; hampered actin reorganization gives rise to abrogated neurite growth and spine formation, which are relevant to the pathophysiology of ID and microcephaly in PLEKHG2-associated neurological disorder. Further intensive analyses of the molecular machinery should contribute to a better understanding of the pathophysiology of the disease.

## 5. Conclusions

The p.Arg204Trp variant in *PLEKHG2* is responsible for postnatal microcephaly and ID. In this study, mouse Plekhg2 was shown to be essential for axon, dendrite, and synapse development in cortical neurons in vivo. Then, pathophysiological significance of the *Plekhg2* variant p.Arg200Trp corresponding to p.Arg204Trp in *PLEKHG2* was examined using a mouse model. As for an underlying mechanism of the pathogenesis, hampered actin reorganization by disrupted PLEKHG2-RAC/CDC42-PAK1 signaling appeared to be a core event. The obtained results indicate that PLEKHG2 has essential roles in brain development, and disruption of its function may cause defects in neuronal development, which lead to ID and microcephaly.

## Figures and Tables

**Figure 1 cells-11-00696-f001:**
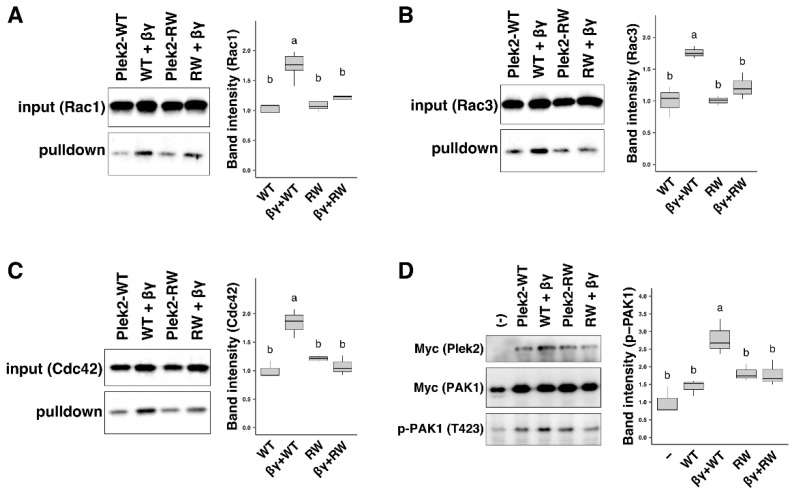
Characterization of the c.610C > T/p.Arg204Trp variation of PLEKHG2 using the mouse PLEKHG2 equivalent Plek2-RW. (**A**–**C**) COS7 cells were co-transfected with pCAG-Μyc-Rac1 (**A**), -Rac3 (**B**), or -Cdc42 (**C**) together with pCAG-Μyc-Plek2-WT, -Plek2-RW, pF5A-CMV-neo-Gβ_1_, and -Gγ_2_ (0.2 μg each), either alone or in combination. Cell lysates were prepared, and the pull-down assay was conducted as described in Section 2 with GST-fused PAK1-PDB (5 μg). The GTP form was indicated by the amounts of each GTPase bound to GST-fused PDB (pulldown). Total cell lysates were also immunoblotted with anti-Myc for normalization (input). Relative band intensity was shown as boxplot when the value of co-expression of Plek2-WT with respective GTPases was taken as 1.0. Number of replicates, N = 3. Different letters above bars represent significant differences, *p* < 0.05, according to a Tukey’s test. (**D**) COS7 cells were transfected with pCAG-Μyc-PAK1, -Plek2-WT, -Plek2-RW, pF5A-CMV-neo-Gβ_1_, and -Gγ_2_ (0.2 μg each), either alone or in combination. PAK1 activation was indicated by the amounts of phosphorylated PAK1 at T423, p-PAK1(T423). Total PAK1 was also immunoblotted with anti-Myc antibody for normalization, Myc (PAK1). Relative band intensity was shown as boxplot when the value of (-) was taken as 1.0. Number of replicates, N = 3. Different letters above bars represent significant differences, *p* < 0.05, according to a Tukey’s test.

**Figure 2 cells-11-00696-f002:**
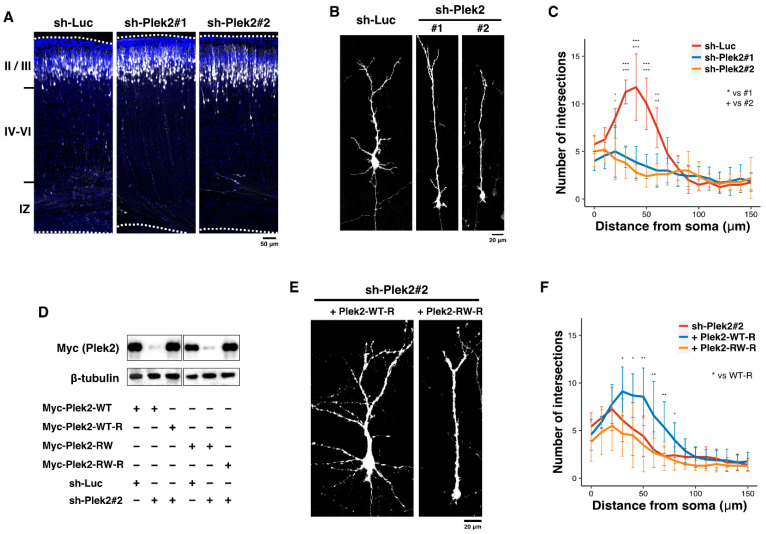
Role of Plekhg2 in the dendrite growth in cortical neurons in vivo. (**A**) pCAG-EGFP (0.5 μg) was co-electroporated in utero with sh-Luc, sh-Plek2#1, or sh-Plek2#2 (1.0 μg each) into the VZ progenitor cells at E14.5. Coronal sections were prepared at P0, and were double-stained with anti-GFP (white) and DAPI (blue). (**B**) Plekhg2-knockdown inhibits dendritic branching in vivo. pCAG-loxP-GFP (0.5 μg) was electroporated for sparse expression with pCAG-M-Cre (0.5 ng) together with sh-Luc, sh-Plek2#1, or sh-Plek2#2 (1.0 μg each) into cerebral cortices at E14.5. Analyses were carried out in cortical slices at P7. Representative average Z-stack projection images of GFP fluorescence of cortical neurons in upper cortical plate were shown. (**C**) Quantification analyses. Branch points of dendrites were analyzed at P7 for sh-Luc, sh-Plek2#1, and sh-Plek2#2 by Sholl test. Error bars indicate +-S.D. More than 8 neurons were analyzed in 2 slices from each of 2 brains obtained from 2 independent electroporation experiments. The significance of difference between control (sh-Luc) and each Plekhg2-knockdown was determined using Dunnett’s test. *** *p* < 0.001, ** *p* < 0.01, * *p* < 0.05, +++ *p* < 0.001, ++ *p* < 0.01, + *p* < 0.05. (**D**) Characterization of RNAi-resistant versions of Plekhg2, Plek2-WT-R and Plek2-RW-R. pCAG-Myc-Plek2-WT, -Plek2-WT-R, -Plek2-RW, or -Plek2-RW-R (0.05 μg each) was transfected into COS7 cells with sh-Luc (Control) or sh-Plek2#2 (1.0 μg each). Western blotting analyses were performed, as shown in Figure 1. (**E**,**F**) pCAG-loxP-GFP (0.5 μg) was electroporated for sparse expression with pCAG-M-Cre (0.5 ng) together with sh-Plek2#2 (1.0 μg) + pCAG-Myc-Plek2-WT-R or -Plek2-RW-R (0.2 μg each) into cerebral cortices at E14.5. Analyses were performed as shown in (**B**,**C**). More than 8 neurons were analyzed in 2 slices from each of 2 brains obtained from 2 independent electroporation experiments. The significance of difference between control (sh-Plek2#2), sh-Plek2#2 + Plek2-WT-R, and sh-Plek2#2 + Plek2-RW-R was determined using Dunnett’s test. ** *p* < 0.01, * *p* < 0.05.

**Figure 3 cells-11-00696-f003:**
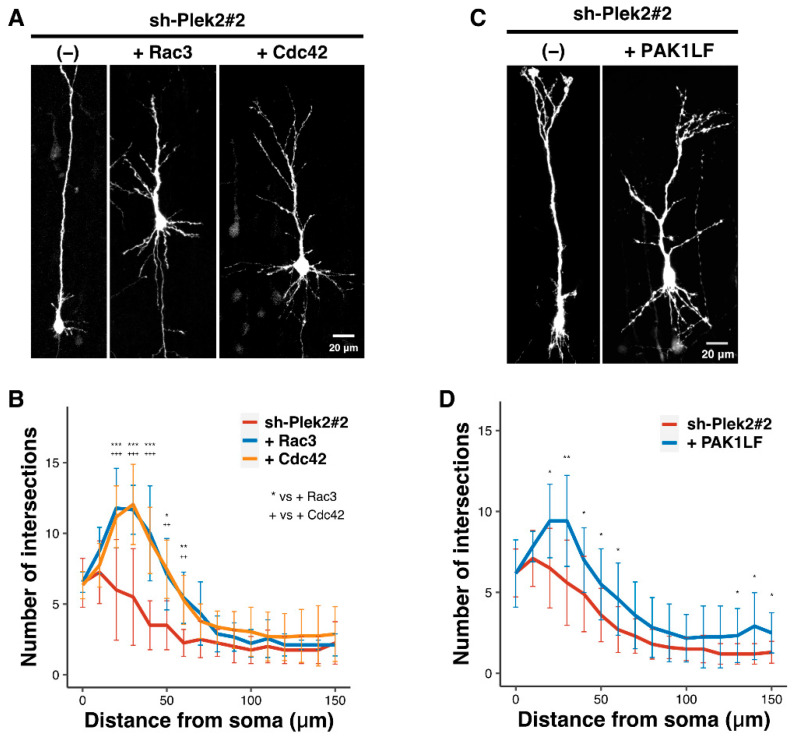
Rescue effects by Rac3, Cdc42, and an active form of PAK1 on the dendrite growth defects caused by Plekhg2-knockdown. (**A**,**C**) pCAG-loxP-GFP (0.5 μg) was electroporated for sparse expression with pCAG-M-Cre (0.5 ng) together with sh-Plek2#2 (1.0 μg) + pCAG-Myc-Rac3, -Cdc42 (0.1 μg each) (**A**), or -PAK1LF (0.2 μg) (**C**) into cerebral cortices at E14.5. Analyses were performed as shown in Figure 2. (**B**) Quantification analyses of (**A**). More than 8 neurons were analyzed in 2 slices from each of 2 brains obtained from two independent electroporation experiments. The significance of difference between control (sh-Plek2#2), sh-Plek2#2 + Rac3, and sh-Plek2#2 + Cdc42 was determined using Dunnett’s test. *** *p* < 0.001, ** *p* < 0.01, * *p* < 0.05, +++ *p* < 0.001, ++ *p* < 0.01. (**D**) Quantification analyses of (**C**). More than 8 neurons were analyzed in 2 slices from each of 2 brains obtained from two independent electroporation experiments. The significance of difference between control (sh-Plek2#2) and sh-Plek2#2 + PAK1LF was determined using Welch’s *t*-test. ** *p* < 0.01, * *p* < 0.05.

**Figure 4 cells-11-00696-f004:**
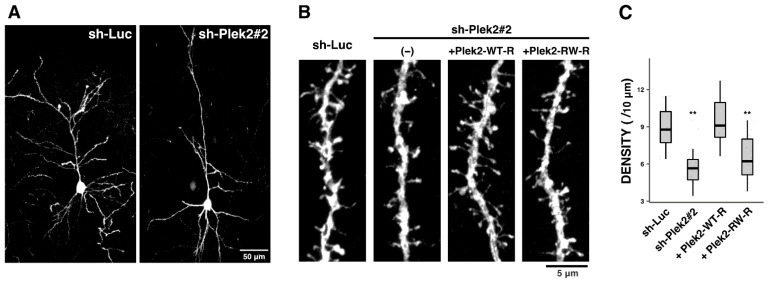
Role of Plekhg2 in the dendritic spine formation in vivo. (**A**,**B**) pCAG-loxP-GFP (0.5 μg) was electroporated for sparse expression with pCAG-M-Cre (0.5 ng) together with sh-Luc or sh-Plek2#2 (1.0 μg each) + none (-), pCAG-Myc-Plek2-WT-R, or -Plek2-RW-R (0.2 μg each) into cerebral cortices at E14.5. Representative average Z-stack projection images of GFP fluorescence of control (sh-Luc) and Plekhg2-deficient neurons (sh-Plek2#2) in upper cortical plate were shown (**A**). Magnified images of the dendritic spines for each experimental condition are shown (**B**). (**C**) Quantitative analyses of density of dendritic spines for each condition in (**B**). More than 8 neurons were analyzed in 2 slices from each of 2 brains obtained from two independent electroporation experiments. The significance of difference between sh-Luc (control) and Plekhg2-knockdown or each rescue condition was determined using Dunnett’s test. ** *p* < 0.01.

**Figure 5 cells-11-00696-f005:**
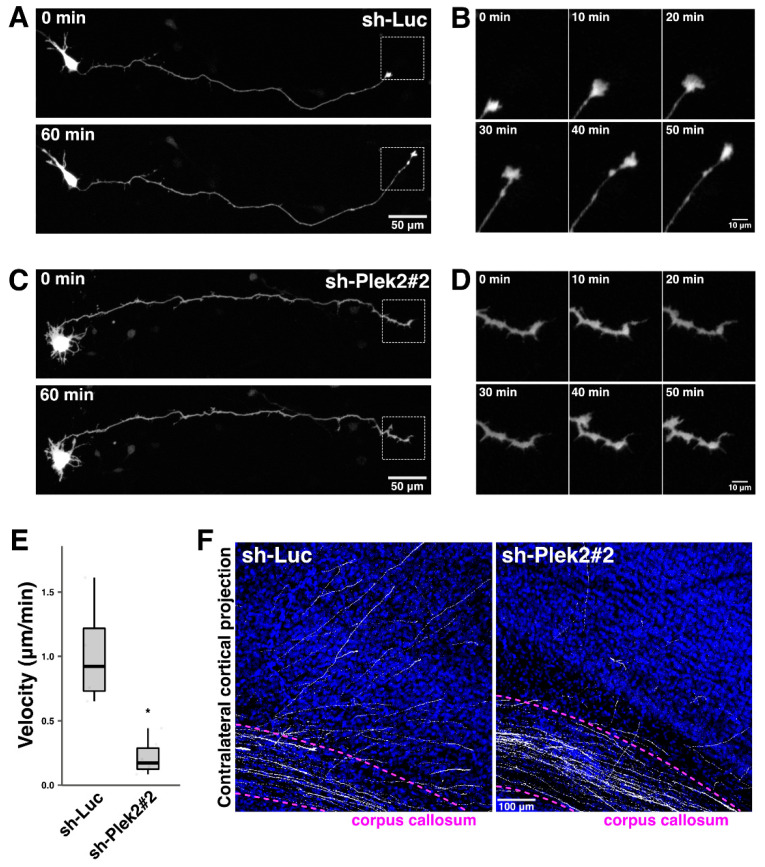
Role of Plekhg2 in the axon growth in cortical neurons in vitro and in vivo. (**A**–**D**) Dissociated cortical neurons from E16 mice were co-electroporated with pCAG-EGFP (0.2 μg) together with sh-Luc (**A**,**B**) or sh-Plek2#2 (**C**,**D**) (1.0 μg each). Cells were observed for 60 min at 2 div. Boxed areas in (**A**,**C**) were magnified in (**B**,**D**), respectively. (**E**) The velocity of growing axon was shown in box plot. The significance of difference between sh-Luc and sh-Plek2#2 was determined using Welch’s *t*-test. * *p* < 0.05. (**F**) Representative images of the terminal arbors of callosal axons expressing GFP with sh-Luc (control) or sh-Plek2#2 (1.0 μg each) at P7. More than 2 slices from each of 2 brains obtained from two independent electroporation experiments were analyzed.

## Data Availability

The data that support the findings of this study are available from the corresponding authors, upon reasonable request.

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
