# Peer review of "Impaired Function of PLEKHG2, a Rho-Guanine Nucleotide-Exchange Factor, Disrupts Corticogenesis in Neurodevelopmental Phenotypes"

_cells, 2022, doi:10.3390/cells11040696_

Round 1
Reviewer 1 Report
Point mutations in the PLEKHG2 RhoGEF are associated with microcephaly associated with intellectual disability, and in this paper the role of this protein on cortical development was assessed. The R204W mutant was found to be less effective at activating Rac1, Rac3 and CDC42 in response to trimeric beta-gamma subunit stimulation. In vivo experiments in which endogenous Plekhg2 was knocked down by RNAi revealed defects in dendrite arbor formation that could be rescued by wild-type Plekhg2 but not the R204W mutant. In addition, the effect of the knockdown could be rescued by Rac3, CDC 42 or active PAK1, consistent with the role of Plekhg2 being activation of this pathway.
Overall the experiments are well designed and convincing, with appropriate statistical tests. The manuscript is clearly written and well organized. One criticism is that there are several experiments for which data were not shown, and given the brevity of the paper and simplicity of the figures, it is strong suggested that at least some of the more important results be shown in the primary figures, even if the results of the experiments indicate lack of effects.
Additional points to be considered.
- In Figure 1, how reproducible are the effects of the mutations on GEF activity?
- Figure 1 apparently shows 2 separate effects of the mutation, some increase in basal GEF activity and reduced response to beta-gamma stimulation for Rac1 and Rac3. In contrast, the basal GEF activity for CDC42 is unaffected and beta-gamma stimulated GEF activity is decreased Are these effects reproducible and significant? What is the hypothesis for these observations? The original hypothesis was that the DH domain mutation would simply effect GEF activity overall, but the data seems to indicate that it is more complicated than that.
- Lines 168-169. It should be clarified in the text (not just the figure legend) describing Figure 1 that the PAK1 was transfected myc-tagged PAK1 that was assayed for T423 phosphorylation.
- Lines 269-270. Why not show the data indicating that knockdown of Plekhg2 had little affect on spine length and head diameter?
- Lines 300-301. Similarly, why not show the data indicating that Plekhg2 silencing did not affect axon bundle extensions into the contralateral hemisphere? Just because data are negative, they may be important and there is room in the primary figures to show them.
- From line 315 forward (Discussion). How do observations regarding the effect of RAC1 mutations relate to PLEKHG2 given that the experiments in this manuscript with PLEKHG2 have examined RAC3 and CDC42 in Figure 3? Particularly since RAC1 was not examined since it had an opposite effect to RAC3 and CDC42?
- Line 338. Again, why not show negative data, in this case results indicating that Plekhg2 knockdown had little effect on neuronal precursor proliferation. If properly controlled, the results are not trivial.
Author Response
Comments to the Authors:
Referee: 1
Point mutations in the PLEKHG2 RhoGEF are associated with microcephaly associated with intellectual disability, and in this paper the role of this protein on cortical development was assessed. The R204W mutant was found to be less effective at activating Rac1, Rac3 and CDC42 in response to trimeric beta-gamma subunit stimulation. In vivo experiments in which endogenous Plekhg2 was knocked down by RNAi revealed defects in dendrite arbor formation that could be rescued by wild-type Plekhg2 but not the R204W mutant. In addition, the effect of the knockdown could be rescued by Rac3, CDC 42 or active PAK1, consistent with the role of Plekhg2 being activation of this pathway.
Overall the experiments are well designed and convincing, with appropriate statistical tests. The manuscript is clearly written and well organized. One criticism is that there are several experiments for which data were not shown, and given the brevity of the paper and simplicity of the figures, it is strong suggested that at least some of the more important results be shown in the primary figures, even if the results of the experiments indicate lack of effects.
Additional points to be considered.
- In Figure 1, how reproducible are the effects of the mutations on GEF activity?
> We agree with the reviewer’s concern which was also raised by the reviewer #2. We repeated the experiments sufficient times to allow calculating the mean activities +/- error of Rac1, Rac3 and Cdc42 activities and Pak1 phosphorylation from several experiments, with valid statistical analysis. Accordingly, we renewed Fig.1 in the new Ms. We also found that, in quantification analyses, Gβγ-mediated activation of Rac1, Rac3 and Cdc42 was inhibited by the mutation. We described the results in the new Ms (line 180 – 183).
- Figure 1 apparently shows 2 separate effects of the mutation, some increase in basal GEF activity and reduced response to beta-gamma stimulation for Rac1 and Rac3. In contrast, the basal GEF activity for CDC42 is unaffected and beta-gamma stimulated GEF activity is decreased Are these effects reproducible and significant? What is the hypothesis for these observations? The original hypothesis was that the DH domain mutation would simply effect GEF activity overall, but the data seems to indicate that it is more complicated than that.
> As pointed out by the reviewer, we repeated the experiments and performed quantitative analyses to determine the effects of the RW mutation on Plekhg2-DH-mediated activation of Rac1, Rac3 and Cdc42. Consequently, we found that Gβγ-mediated activation of Rac1, Rac3 and Cdc42 was inhibited by the mutation. Wild type Plekhg2-DH caused activation of all the 3 GTPases when co-expressed with Gβγ, whereas the mutated Plekhg2-DH did not statistically activate Rac1, Rac3 and Cdc42. As for the Plekhg2-DH-mediated PAK1-phosphorylation (activation) was also lost when the RW mutant was used. These results suggest that Rac1, Rac3 and Cdc42 are involved in PAK1 signaling in this transient expression system. Based on the obtained results, we now consider that the DH domain mutation is simply effective in GEF activity overall. We described this issue in the new Ms (line 180 – 185). We would appreciate the valuable comment by the reviewer.
- Lines 168-169. It should be clarified in the text (not just the figure legend) describing Figure 1 that the PAK1 was transfected myc-tagged PAK1 that was assayed for T423 phosphorylation.
> According to the Reviewer’s suggestion, we clarified in the text that the T423 phosphorylation of PAK1 was conducted with transfected myc-tagged PAK1 in the new Ms (line 183 – 185).
- Lines 269-270. Why not show the data indicating that knockdown of Plekhg2 had little affect on spine length and head diameter?
> According to the Reviewer’s suggestion, we showed the data indicating that knockdown of Plekhg2 had little effect on spine length and head diameter as a Supplementary Figure 1 in the new Ms (line 301). We also described the method in the supplementary information in the new Ms.
- Lines 300-301. Similarly, why not show the data indicating that Plekhg2 silencing did not affect axon bundle extensions into the contralateral hemisphere? Just because data are negative, they may be important and there is room in the primary figures to show them.
> According to the Reviewer’s suggestion, we showed the data indicating that knockdown of Plekhg2 had little effect on axon bundle extensions into the contralateral hemisphere as a Supplementary Figure 2 in the new Ms (line 335). We also described the method in the figure legend in the new Ms (Supplementary information in the new Ms).
- From line 315 forward (Discussion). How do observations regarding the effect of RAC1 mutations relate to PLEKHG2 given that the experiments in this manuscript with PLEKHG2 have examined RAC3 and CDC42 in Figure 3? Particularly since RAC1 was not examined since it had an opposite effect to RAC3 and CDC42?
> As the reviewer pointed out, we actually tried to observe the effects of Rac1. However, since overexpression of wild type Rac1 (0.1 µg electroporation) caused drastic migration defects, it was difficult to find neurons distributed in layer II/III of cerebral cortex. In contrast, Rac3 and Cdc42 had little effects on the cortical neuron migration under the same experimental conditions, and thus we could examine the dendritic morphology. Although we tried to determine the proper experimental conditions, it was difficult because even as little amount as 0.03 µg electroporation of Rac1 caused severe migration defects when electroporated. We would appreciate it if the Reviewer could kindly consider and accept the above explanation.
- Line 338. Again, why not show negative data, in this case results indicating that Plekhg2 knockdown had little effect on neuronal precursor proliferation. If properly controlled, the results are not trivial.
> According to the Reviewer’s suggestion, we displayed the data indicating that knockdown of Plekhg2 had little effect on neuronal precursor proliferation as a Supplementary Figure 3 in the new Ms (line 372). We also described the method in the supplementary information in the new Ms.
Reviewer 2 Report
Plekhg2 (also known as Clg) is an interesting yet understudied Rho-GEF that interacts directly with the Gbg subunits of heterotrimeric G proteins. Coexpression of Plekhg2 and Gbg leads to increased activity of the Small GTPases Rac and Cdc42 in various cell types. In humans, a point mutation in Plekhg2 (R204W) causes microcephaly.
Here, the authors investigate the role of Plekhg2 in neuronal development using shRNA-mediated knock-down of Plekh2 in mice by in-utero electroporation, as well as evaluating the mouse equivalent of the human disease, Plekhg2-R200W.
Overexpression in Cos7 cells suggests that Plekhg2-R200W may have lost the ability to activate Rac1, Rac3 and Cdc42 when co-expressed with Gbg. These data need firming up.
Knockdown of Plekhg2 in mice did not affect the migration of cortical neurons in the developing embryo, but delayed dendritic arbour formation and dendritic spine density, as well as axon pathfinding. Dendritic arbour formation and spine density could be rescued by wild-type Plekhg2 but not Plekhg2-R200W, suggesting that R200W is a loss-of-function mutant. Dendritic arbour development could also be rescued by Rac3, Cdc42 and active Pak1, suggesting that Plekhg2 may exert its effect on cortical neuron development through its catalytic GEF activity.
The paper is very interesting. However, there are several over-interpretations of data, and additional experiments will be required.
Major:
1) Fig1: Wild type Plekhg2 or R200W-Plekhg2 were co-transfected with Gbg into Cos7 cells, and Rac1, Rac3 and Cdc42 activities were evaluated by pull down assays, as well as the phosphorylation of the Rac and Cdc42 target Pak1. The data seem to indicate that co-transfection of the wild type but not Plekhg2-R200W mutant with Gbg leads to increased activities of these GTPases and phosphorylation of Pak1.
However, relative band intensities are given only for the blots shown. These experiments need to be quantified properly. The assays must be repeated sufficient times to allow calculating the mean activities +/- error of Rac1, Rac3 and Cdc42 activities and Pak1 phosphorylation from several experiments, with valid statistical analysis.
Furthermore, the pull down assays were done with overexpressed GTPases. The authors must mention this clearly in the Results text and emphasize that therefore the results may not reflect the physiologically relevant in vivo substrate specificity of Plkhg2, as this can be overridden by the overexpression of GTPases (e.g. Rac3 is not expressed in Cos7 cells).
2) Figs 2-4 and 5F: the defect in dendrite formation and spine density upon Plekhg2 knockdown and the rescue experiments look very convincing. Please specify in the figure legends the number of embryos tested in each case, from how many independent in utero electroporations, how many slices per brain were analysed, and how many cells per slice. For Figs 5A-E, please indicate the number of cells analysed.
3) Introduction, line 40: The authors state that Plekhg2 (is a Gbg-activated GEF. I have not seen any evidence that proves this. Plekhg2 was indeed shown to interact directly with Gbg in vitro, and also to interact with Gbg in vivo, through a stretch directly N-terminal of DH domain (Ueda H et al JBC 283:1946). Furthermore, coexpression of Plekhg2 and Gbg leads to increased Rac and Cdc42 (but not RhoA activity) in various cell types. However, the latter is no proof that Gbg directly activate the GEF activity of Plekhg2, as they do for example with Prex Rac-GEFs. To prove that, in vitro GEF assays with recombinant Plekhg2 and Gbg proteins would be required. Co-expression could simply reflect indirect activation via one or more additional factors.
The authors need to tone down the text accordingly to state no more than Plekhg2 binds Gbg directly and is activated upon co-expression with Gbg in vivo.
Minor:
4) The manuscript should be edited by English language scientific writing experts, there are mistakes in phrasing throughout. Although the meaning is clear, this makes reading somewhat unpleasant.
5) Materials and Methods, Section 2.1 Ethics statement: Specify the housing conditions (cage types, lighting cycles, food), strain sex, age-range and pathogen-status of the mice used.
6) Line 82, please indicate ‘the late’ Prof Alan Hall
7) The International Mouse Phenotyping Consortium has phenotyped a mouse strain with global Plekhg2 deficiency, and out of 75 parameters assessed, the only significant defect they found was decreased body length.
https://www.mousephenotype.org/data/genes/MGI:2141874
Can the authors please comment in the Discussion how this phenotype might be reconciled with their dataset.
8) Line 238 cites submitted papers. Is that permitted by the journal?
9) Fig 5F appears too dark on my computer. Can be photos be made brighter?
Author Response
Plekhg2 (also known as Clg) is an interesting yet understudied Rho-GEF that interacts directly with the Gbg subunits of heterotrimeric G proteins. Coexpression of Plekhg2 and Gbg leads to increased activity of the Small GTPases Rac and Cdc42 in various cell types. In humans, a point mutation in Plekhg2 (R204W) causes microcephaly.
Here, the authors investigate the role of Plekhg2 in neuronal development using shRNA-mediated knock-down of Plekh2 in mice by in-utero electroporation, as well as evaluating the mouse equivalent of the human disease, Plekhg2-R200W.
Overexpression in Cos7 cells suggests that Plekhg2-R200W may have lost the ability to activate Rac1, Rac3 and Cdc42 when co-expressed with Gbg. These data need firming up.
Knockdown of Plekhg2 in mice did not affect the migration of cortical neurons in the developing embryo, but delayed dendritic arbour formation and dendritic spine density, as well as axon pathfinding. Dendritic arbour formation and spine density could be rescued by wild-type Plekhg2 but not Plekhg2-R200W, suggesting that R200W is a loss-of-function mutant. Dendritic arbour development could also be rescued by Rac3, Cdc42 and active Pak1, suggesting that Plekhg2 may exert its effect on cortical neuron development through its catalytic GEF activity.
The paper is very interesting. However, there are several over-interpretations of data, and additional experiments will be required.
Major:
1) Fig1: Wild type Plekhg2 or R200W-Plekhg2 were co-transfected with Gbg into Cos7 cells, and Rac1, Rac3 and Cdc42 activities were evaluated by pull down assays, as well as the phosphorylation of the Rac and Cdc42 target Pak1. The data seem to indicate that co-transfection of the wild type but not Plekhg2-R200W mutant with Gbg leads to increased activities of these GTPases and phosphorylation of Pak1.
However, relative band intensities are given only for the blots shown. These experiments need to be quantified properly. The assays must be repeated sufficient times to allow calculating the mean activities +/- error of Rac1, Rac3 and Cdc42 activities and Pak1 phosphorylation from several experiments, with valid statistical analysis.
> We agree with the reviewer’s concern about proper quantification of the presented data. We repeated the experiments 3 times to allow valid statistical analysis with Tukey’s test. Accordingly, we renewed Fig.1 in the new Ms and altered the description in the revised Ms (line 198 – 200, line 204-206).
Furthermore, the pull down assays were done with overexpressed GTPases. The authors must mention this clearly in the Results text and emphasize that therefore the results may not reflect the physiologically relevant in vivo substrate specificity of Plkhg2, as this can be overridden by the overexpression of GTPases (e.g. Rac3 is not expressed in Cos7 cells).
> We agree with the reviewer’s comment. We clarified in the Results section that the pull down assays were done with overexpressed GTPases and thus the results may not reflect the physiologically relevant in vivo substrate specificity of Plkhg2 in the new Ms (line 175 – 177, line 187 – 189, line 401, line 404 – 406).
2) Figs 2-4 and 5F: the defect in dendrite formation and spine density upon Plekhg2 knockdown and the rescue experiments look very convincing. Please specify in the figure legends the number of embryos tested in each case, from how many independent in utero electroporations, how many slices per brain were analysed, and how many cells per slice. For Figs 5A-E, please indicate the number of cells analysed.
> According to the reviewer’s comment, we specified in the figure legends of Figs 2-4 and 5F the number of embryos tested in each case, from how many independent in utero electroporations, how many slices per brain were analysed, and how many cells per slice in the new Ms. Also, we indicated the number of cells analysed in Figs 5A-E in the new Ms.
3) Introduction, line 40: The authors state that Plekhg2 (is a Gbg-activated GEF. I have not seen any evidence that proves this. Plekhg2 was indeed shown to interact directly with Gbg in vitro, and also to interact with Gbg in vivo, through a stretch directly N-terminal of DH domain (Ueda H et al JBC 283:1946). Furthermore, coexpression of Plekhg2 and Gbg leads to increased Rac and Cdc42 (but not RhoA activity) in various cell types. However, the latter is no proof that Gbg directly activate the GEF activity of Plekhg2, as they do for example with Prex Rac-GEFs. To prove that, in vitro GEF assays with recombinant Plekhg2 and Gbg proteins would be required. Co-expression could simply reflect indirect activation via one or more additional factors.
The authors need to tone down the text accordingly to state no more than Plekhg2 binds Gbg directly and is activated upon co-expression with Gbg in vivo.
> We agree with the reviewer’s concern that Gbg may not directly activate the GEF activity of Plekhg2 and co-expression could simply reflect indirect activation via one or more additional factors. We thus tone down the text and stated that Plekhg2 interacts with Gbg directly and is activated upon co-expression with Gbg in cellular context but direct activation by Gbg remains to be unknown (line 42 – 45 in the new Ms).
Minor:
4) The manuscript should be edited by English language scientific writing experts, there are mistakes in phrasing throughout. Although the meaning is clear, this makes reading somewhat unpleasant.
> According to the suggestion, we asked Prof. Lynne E. Maquat at University of Rochester language editing. She kindly read throughout the Ms and improved the language. We put this in the “Acknowledge” section in the new Ms.
5) Materials and Methods, Section 2.1 Ethics statement: Specify the housing conditions (cage types, lighting cycles, food), strain sex, age-range and pathogen-status of the mice used.
> According to the reviewer’s comment, we specified the housing conditions (cage types, lighting cycles, food), strain sex, age-range and pathogen-status of the mice used in the new Ms (line 128 – 130).
6) Line 82, please indicate ‘the late’ Prof Alan Hall
> We agree to the reviewer’s and changed the expression.
7) The International Mouse Phenotyping Consortium has phenotyped a mouse strain with global Plekhg2 deficiency, and out of 75 parameters assessed, the only significant defect they found was decreased body length.
https://www.mousephenotype.org/data/genes/MGI:2141874
Can the authors please comment in the Discussion how this phenotype might be reconciled with their dataset.
> We would appreciate the reviewer’s kind suggestion. According to the comment, we added description on the differences between our phenotypes and those obtained in the KO mouse in the new Ms (line 387 - 395). We would appreciate it if the Reviewer could kindly accept the above description.
8) Line 238 cites submitted papers. Is that permitted by the journal?
> We agree with the reviewer’s concern and thus deleted the 2 papers under submission in the new Ms.
9) Fig 5F appears too dark on my computer. Can be photos be made brighter?
> We are sorry for the inconvenience. We changed the pictures to brighter ones in the new Ms.
Round 2
Reviewer 1 Report
The authors have satisfactorily addressed the issues raised in the first review.
Reviewer 2 Report
The authors have addressed all my previous concerns in the revised manuscript.